



Earth **Surface** Dynamics

Discussions

# Dominant process zones in a mixed fluvial-tidal delta are morphologically distinct

Mariela Perignon[1], Jordan Adams[2], Irina Overeem[2,3], and Paola Passalacqua[1]

[1]Department of Civil, Architectural and Environmental Engineering, Cockrell School of Engineering, The University of Texas at Austin, Austin, Texas, USA
[2]Community Surface Dynamics Modeling System (CSDMS), Institute of Arctic and Alpine Research, University of Colorado at Boulder, Boulder, Colorado, USA
[3]Department of Geological Sciences, University of Colorado at Boulder, Boulder, Colorado, USA

**Correspondence:** Paola Passalacqua (paola@austin.utexas.edu)

**Abstract.** The morphology of deltas is determined by the spatial extent and variability of the geomorphic processes that shape them. While in some cases resilient, deltas are increasingly threatened by natural and anthropogenic forces, such as sea level rise and land use change, which can drastically alter the rates and patterns of sediment transport. Quantifying process patterns can improve our predictive understanding of how different zones within delta systems will respond to future change. Available

remotely sensed imagery can help but appropriate tools are needed for pattern extraction and analysis. We present a method for extracting information about the nature and spatial extent of active geomorphic processes across deltas from ten parameters quantifying the geometry of each of 1,239 islands and the channels around them using machine learning. The method consists of a two-step unsupervised machine learning algorithm, that clusters islands into spatially continuous zones based on the ten morphological metrics extracted from remotely sensed imagery. By applying this method to the Ganges-Brahmaputra-

Meghna Delta, we find that the system can be divided into six major zones. Classification results show that active fluvial island construction and bar migration processes are limited to relatively narrow zones along the main Ganges River and Brahmaputra and Meghna corridors, whereas zones in the mature upper delta plain, with smaller fluvial distributary channels stand out as their own morphometric class. The classification also shows good correspondence with known gradients in the influence of tidal energy with distinct classes for islands in the backwater zone and in the purely tidally-controlled region of the delta.

Islands at the delta front, under the mixed influence of tides, fluvial-estuarine construction, and local wave reworking have their own characteristic shape and channel configuration. The method does not distinguish between islands with embankments (polders) and natural islands in the nearby mangrove forest (Sundarbans), suggesting that human modifications have not yet altered the gross geometry of the islands beyond their previous 'natural' morphology. These results demonstrate that machine learning and remotely sensed imagery are useful tools for identifying the spatial patterns of geomorphic processes across delta

systems.



# 1 Introduction

Deltaic environments are threatened by environmental change and anthropogenic activity. Predicting the response of deltas to these future changes requires understanding the spatial variability of physical processes and their influence on landscape morphology. The identification of the patterns of physical processes in deltas is therefore necessary to predict the resilience of these environments and assure the sustainable use of the environmental services they provide. It has long been established that the morphology of deltaic networks and landforms reflects the physical processes that created and continuously modified them (e.g., Galloway, 1975). Although relationships between specific geomorphic processes, island morphology, and channel geometry have been analyzed in less complex deltaic systems (Smart and Moruzzi, 1971; Edmonds et al., 2011), isolating the effects of individual physical processes on morphology is challenging in large deltas where multiple processes interact or where the relative influence of each process can change over time. Here we propose the use of machine learning techniques to identify the spatial patterns of geomorphic processes based on the morphology of islands, their internal drainage networks, and the channels that bound them.

The morphology of deltas is set by interactions and feedbacks between water and sediment fluxes throughout the system (Orton and Reading, 1993; Edmonds and Slingerland, 2010), including the physical processes that transport them across the landscape (Wright and Coleman, 1972; Galloway, 1975) and the boundary conditions that determine their pathways (Orton and Reading, 1993; Syvitski and Saito, 2007). Many classification schemes have been proposed to qualitatively connect the geometry of deltas to geomorphic processes. Early work by Wright and Coleman (1972) and Galloway (1975) used the plan-view geometry of deltas to classify deltaic systems according to the relative influence of fluvial input, tidal currents, and wave energy on their morphology. Others have expanded this classification scheme to account for the effects of sediment grain size (Postma, 1990; Orton and Reading, 1993; Caldwell and Edmonds, 2014), sediment cohesion (Edmonds and Slingerland, 2010), and base level change (Postma, 1990; Dalrymple et al., 1992; Wolinsky et al., 2010). Common relationships between delta-scale morphological metrics and key factors that control delta morphology have been found at a global scale (Syvitski et al., 2005; Syvitski and Saito, 2007).

The classification of deltas according to their bulk characteristics provides limited information about the geomorphic processes that locally modify each element of the system (Edmonds et al., 2011). Most deltaic environments display complex morphologies that result from spatial variability in physical processes (Restrepo et al., 2002; Syvitski and Saito, 2007; Lewin and Ashworth, 2014). Metrics of channel morphology have been widely used to characterize the influence of external forcings acting in different parts of a delta. Fagherazzi et al. (1999) and Rinaldo et al. (1999) found that scaling relationships and topology of tidal channel networks, unlike fluvial systems, can vary significantly among tidal basins due to the influence of multiple physical processes. In mixed fluvial-tidal systems, an along-channel break in geometric scaling relationships has been found at the point where the influence of tides on channel morphology becomes stronger than the river influence (Sassi et al., 2012; Kästner et al., 2017).

The processes that dominate delta morphology may change over time as delta shape evolves or environmental conditions change (Correggiari et al., 2005). By looking at the evolution of channel planform overlap, morphological metrics have been



used to quantify growth (Wolinsky et al., 2010) and channel network dynamics (Cazanacli et al., 2002; Liang et al., 2016b) over time. Channel networks can also preserve information about their evolution; in fluvially-dominated deltas, Jerolmack and Swenson (2007) identified differences in channel and network morphology for distributary systems that evolve through mouth-bar deposition and those formed by avulsions.

Approaches used to quantify spatial patterns in geomorphic processes have also relied on mathematical descriptions of the system as a network (Passalacqua, 2017), which are helpful not only for the analysis of network structure and dynamics but also for the quantification of connectivity between channels and islands (Hiatt and Passalacqua, 2015). For example, Trigg et al. (2012) identified reaches along the Amazon River that were morphologically distinct and had mostly separate connectivity networks, suggesting spatial differences in hydrology. In the Jamuna River in Bangladesh, Marra et al. (2014) used a centrality

property of the channel network to quantify the importance of individual strands of the braided river and capture changes in linkages over time. Tejedor et al. (2015a, b) developed a quantitative framework for studying delta channel networks and the propagation of perturbations using spectral graph theory (Tejedor et al., 2016). Spatial variability in the morphology of the channel network also results in diversity in the geometry of islands across a delta (Meshkova and Carling, 2013), that have been quantified with multiple metrics of island and network morphology (Edmonds et al., 2011), with statistical analyses

(Passalacqua et al., 2013), and in numerical models under various input conditions, sea level rise, and subsidence (Liang et al., 2016b, a).

    The use of machine learning in earth surface sciences is rapidly increasing as the volume and complexity of available data grows (e.g., Rubin, 1992; Jaffe and Rubin, 1996; Werner, 1999; Murray et al., 2009, 2014; Goldstein et al., 2019). Machine learning algorithms generalize large samples of observations to identify and exploit patterns in the data. These techniques

are traditionally divided into supervised methods, where the system is trained to identify patterns based on a set of samples with known labels, and unsupervised methods, where classes in the dataset have not been previously identified. Machine learning techniques are used extensively to address a broad range of problems where grouping in data are not immediately obvious or where the ability to predict behavior is required. Valentine and Kalnins (2016) and Goldstein et al. (2019) present comprehensive overviews of the use of machine learning in the geosciences. Applications span a wide range of topics, such

as: streamflow modeling and forecasting (Asefa et al., 2006; Rasouli et al., 2012; Shortridge et al., 2016), runoff modeling (Gudmundsson and Seneviratne, 2015), flood risk assessment (Dibike and Solomatine, 2001; Tehrany et al., 2014; Mojaddadi et al., 2017), sediment yield variability (Tamene et al., 2006), and sediment transport (Bhattacharya et al., 2007; Melesse et al., 2011; Schmelter et al., 2011; Choubin et al., 2018). Particularly relevant to the analysis of river networks is the work on surface water extraction (Pekel et al., 2016; Donchyts et al., 2016; Isikdogan et al., 2017a, 2019) and on delta network extraction

(Isikdogan et al., 2018).

    In this paper, we propose a two-step unsupervised machine learning method to analyze spatial patterns in large river deltas. We apply this method to the Ganges-Brahmaputra-Meghna Delta and show that by clustering areas of the distributary system with common morphological characteristics, we are able to reliably extrapolate our understanding of the physical processes that locally dominate island and channel morphology across large areas of the delta, greatly increasing the explanatory power of on-

the-ground observations or high-resolution numerical modeling. This hierarchical clustering of islands in the delta according



to their common characteristics also allows us to identify areas of the landscape that would be affected by different forecasted scenarios of future environmental conditions.

## 2 Case study: The Ganges-Brahmaputra-Meghna Delta

The Ganges-Brahmaputra-Meghna Delta (GBMD) covers more than 100,000 km$^2$ of Bangladesh and eastern India, extending
~400 km from its apex near the foothills of the Himalaya to the Bay of Bengal (Wilson and Goodbred, 2015). The largest subaerial delta in the world, the GBMD system has been influenced by tectonic, climate, fluvial and tidal forcings. Most of Bangladesh is underlain by alluvial deposits, sourced from the uplifting Himalaya orogen and deposited into accommodation space created by local subsidence in the basin (Allison, 1997; Allison et al., 2003; Kuehl et al., 2005). Most of the region lies within 20 m of mean sea level, and local relief is minimal. In the uppermost delta, the Ganges and Brahmaputra rivers formed
a fan delta that has slopes of ~10$^{-4}$. The tidal plains, lying at or near sea level, have lower surface slopes, ~10$^{-5}$ (Wilson and Goodbred, 2015).

Fluvial processes control the morphology of the eastern portion of the delta. The Ganges and Brahmaputra rivers are sediment-laden. Together they transport about 10$^9$ tons of sediment per year to the Bay of Bengal and form a broad and elevated braid plain extending 5-15 km from the present-day active rivers (Goodbred and Kuehl, 2000; Wilson and Goodbred,
2015). Bars and channels evolve rapidly within the braid belts during high discharge events, and stabilize once they are colonized by vegetation and amalgamate to form island complexes (Best et al., 2007; Wilson and Goodbred, 2015). Major avulsions of the Ganges and Brahmaputra rivers occur every 1,500 - 2,500 years (Allison et al., 2003; Pickering et al., 2014; Reitz et al., 2015). Backfilling and underfit meandering streams of the formerly active braid plains form the central and western portions of the upper deltaic plains. This region receives minimal fluvial discharge from the active braid plain to the north and east.

Away from the river-mouth estuary, the tidal plains are formed by dense networks of funnel-shaped tidal channels that receive no upstream input of fluvial sediment. Strong tides, however, generate an onshore flux of suspended sediment originating from the estuary. Population density in much of the lower tidal delta plain is high and several islands were embanked for agriculture in the 1960s and 1970s (Auerbach et al., 2015; Rahman and Salehin, 2013). These embanked land masses, called 'polders', have drastically reduced overbank sedimentation due to natural ebb-flood tidal cycles and are sediment starved relative to
nearby natural islands (Auerbach et al., 2015; Wilson et al., 2017). The construction of embankments for agriculture within the tidal plain has been shown to contribute to the amplification of the tidal signal (Pethick and Orford, 2013). The infilling of small tidal channels in this inhabited region leads to new, or 'khas' land development (Wilson et al., 2017).

The Sundarbans mangrove forest is located to the west and south of the poldered area. Relatively pristine, and covering 4,100 km$^2$ of the western portion of the tidal plains, the Sundarbans continue to aggrade through the reworking and settling of
tidal sediments. Due to sediment deposition, the elevation of islands in the Sundarbans can exceed the elevation of the nearby polders by ~1 m (Auerbach et al., 2015).

Climate in the GBMD is sub-tropical and dominated by the South-East Asian monsoon, resulting in strongly seasonal fluvial discharges (Islam et al., 1999; Goodbred et al., 2003). Monsoonal rains between June and September are the primary source of





runoff to the Brahmaputra and Ganges rivers (Best et al., 2007; Singh, 2007). Approximately 80% of the total yearly sediment load is transported due to monsoonally-driven increases in fluvial discharge (Goodbred and Kuehl, 2000). The southern portion of the delta front is characterized by wide channels, due to the strong diurnal and mesotidal influence in the region (Goodbred and Kuehl, 2000) (Figure 1). There is a change in tidal magnitudes across the delta front: in the Meghna Estuary, tidal amplitude exceeds 5 m. Moving westward across the delta front, mean tidal amplitude decreases to 1.9 m (Allison, 1998).

## 3 Methods

Our goal is to regionalize the study area into zones with common physical characteristics to differentiate the areas of influence of various physical processes. Regionalization attempts to aggregate spatial units or observations into clusters based on spatial continuity as well as attribute similarity (e.g., Guo, 2008; Duque et al., 2012b; Wu et al., 2013). By identifying clusters of islands in the GBMD with shared geomorphic characteristics generated by similar physical processes, we can extrapolate local observations to larger areas of the delta as well as begin to predict how changes in the geomorphic drivers due to natural and anthropogenic forcings might affect delta morphology.

Each step in this methodology is explained further in the following sections: (1) data are first extracted from remotely-sensed imagery and morphometrics are calculated; (2) metrics are normalized and correlation between metrics is addressed with a principal component analysis; (3) islands are clustered into groups; and (4) clusters are grouped and ordered based on a nested, or hierarchical, clustering scheme.

### 3.1 Metrics and data used

Borrowing from computer vision (e.g., Dryden et al., 1997) as well as previous research in surface processes (e.g., Edmonds et al., 2011; Passalacqua et al., 2013), we identified a set of metrics (Fig. 2) that capture the shape of deltaic islands, their internal drainage networks, and the channels that bound them. Metrics used to study delta morphology commonly take a "sediment-focused" approach (Edmonds et al., 2011), based on the idea that sediment dispersal across the system is the primary driver of morphodynamics. To avoid unintentionally encoding geographic information into the feature matrix, we avoided metrics related to the position of islands in the delta (e.g. distance from delta apex, distributary number). Similar methods have been used before to characterize the shape of features such as lakes, icebergs, and streamlined islands in remote sensing imagery (Baker and Kochel, 1979; Komar, 1983; Kehew and Lord, 1986; Vila and Machado, 2004; Frohn et al., 2005; Silva and Bigg, 2005).

We obtained the channel network and water surface mask from Orthorectified Landsat Thematic Mapper Mosaics (28.5 m resolution) as in Passalacqua et al. (2013). Typically, the delta is cloud-covered for much of the monsoonal rainy season, so that composite imagery generally is representative for the dry season state of the delta. A map of interchannel islands can be derived from the water surface map as the land masses bounded by channels. Performing this operation on the GBMD results in 1,239 unique features.





## 3.2 Feature normalization and principal component analysis

We normalized metrics by using a logarithmic normalization to differentiate between islands in the distribution and we then scaled them from 0 to 1 (Fig. 3). Because the subset of metrics selected for this study shows moderate degrees of correlation (Fig. 4), principal component analysis (PCA) is used to convert metrics into parameters that are uncorrelated from one

another. PCA is a dimensionality reduction tool that preserves the variance within the data while eliminating colinearity between features. When too many features are used, the tendency of machine learning is to overfit the data. PCA reduces this dimensionality, while preserving 90% of the variance in the original dataset.

## 3.3 Spatial clustering

Clustering is an application of machine learning where large volumes of multi-dimensional data are reduced into groups of

objects with similar properties (Jain et al., 1999; Fisher, 1987). Spatial clustering is a major challenge for geographic data analysis and increasingly of interest to the machine learning community (Duque et al., 2012a; Gehlke and Biehl, 1934; Guo, 2008; Openshaw et al., 1979). These algorithms are appropriate for applications that require spatially contiguous clusters that contain regions as homogeneous as possible (within each cluster), separated from each other by discrete boundaries. Examples are the creation of areas for precision farming (Fleming et al., 2004) and estuarine management areas (Bação et al., 2005a).

Spatial clustering is based on the idea that objects that are close to each other are more likely to be similar to one another than to objects that are farther away (Tobler, 1969).

Self-Organizing Maps (SOM) are a type of artificial neural network that is widely used for visualization and analysis of high-dimensional non-spatial data (Kohonen, 2001). SOMs reduce high-dimensional data onto an often 2-dimensional grid of nodes and map the input data onto the grid while preserving topological relations between samples. As a result, objects that

are close to one another in parameter space are mapped to nearby nodes on the grid (Kohonen, 2001). While training the SOM, the algorithm iteratively deforms the grid to best fit the n-dimensional parameter space. Initially, each node on the grid takes a random value for each parameter in the input dataset. In each successive iteration, a SOM calculates the Euclidean distance in parameter space between each object in the input dataset and the parameter values for each unit in the grid and assigns the data point to the closest node, called Best Matching Unit. Once all data points are assigned to a node, the parameter values for

nodes in the grid are updated. With increasing iterations, the match between input data points and nodes in the grid improves (Haykin and Principe, 1998).

SOMs have been adapted to solve spatial clustering problems (Agarwal and Skupin, 2008). The GeoSOM algorithm adapts self-organizing maps to consider the geographic distribution of objects when searching for a Best Matching Unit (BMU) (Bação et al., 2005b, 2008). Each node in a GeoSOM grid is in a fixed geographic location within the spatial extent of the input

data. At each iteration, the algorithm first identifies a subset of nodes in the grid that are within a given geographic distance of an object in the input dataset, and then searches for a BMU for that object from only that subset of nodes (Feng et al., 2014). The relative importance of geographic proximity and input parameter values for classification is therefore dependent on





a user-defined geographic tolerance (Bação et al., 2004). This operation results in clusters of data points that are close in both parameter and geographic space.

We adapted and expanded the implementation of GeoSOM in the ClusterPy Python library to improve the search for potential BMUs when the size of objects in the input dataset is comparable to the spacing of the GeoSOM grid. In the case of objects

much larger than individual nodes in the grid, limiting the search for candidate BMUs to those within a certain distance of the object's centroid is likely to assign the object to a node that is within the outline of the object itself, decreasing the likelihood that a large object will be clustered with other objects. To better handle datasets with objects of multiple sizes, we modified the search algorithm to consider as candidates for BMU all nodes that directly intersect the object as well as their neighboring nodes, expanding to further neighbors if a higher geographic tolerance is desired. The resulting algorithm is therefore able to

group large objects with other nearby objects if they are similar in parameter space.

The number of output neurons in the neural network (the grid size) affects the quality of the clustering results. A coarse grid (too few neurons) leads to clusters that are too general and reduces the ability to find significant differences between them. A grid that is very fine (too many neurons) overfits the input data and results in too many clusters that do not generalize variability in the sample (Park et al., 2004; Céréghino and Park, 2009). Although there is no established method for selecting the optimal

number of neurons for a particular classification problem, Vesanto et al. (2000) proposed that the number of nodes in a SOM should be $5\sqrt{n}$, where n is the number of samples. For this study, the number of samples is given by the number of islands in the domain (n = 1,239), suggesting that 176 is the optimal number of nodes. Given the geographic constraints placed on clusters by GeoSOM and the differences between the shape of the domain and the grid, we selected a larger grid of 40 neurons per side (200 neurons) as appropriate for this problem.

## 3.4   Hierarchical Agglomerative Clustering

Hierarchical agglomerative clustering builds nested clusters, starting with treating each observation as a separate cluster. Iteratively, hierarchical agglomerative clustering then executes the following two steps: identifies clusters that are closest together and merges the two most similar clusters according to a measure of dissimilarity. This process repeats until all clusters are merged together, creating a classification of all islands (Fig. 5). The hierachy of clusters is represented as a 'tree' or dendro-

gram (Fig. 6). The 'root' of the 'tree' is the unique cluster that gathers all samples, and the 'leaves' are the remaining clusters with only one sample. The most common metric of dissimilarity between clusters is Ward's linkage method. Ward's method is used to minimize variance within a hierarchical approach. Variance minimization serves as a threshold that stops clusters from grouping together. To enforce the formation of geographically contiguous clusters, connectivity constraints were imposed on the agglomerative clustering algorithm so that only adjacent nodes in the GeoSOM grid could be merged together. Through

this operation, the delta islands (Fig. 5) can be clustered into their adjacent groups (Figure 7).





## 4   Results

### 4.1   Metric relationships: feature normalization and principal component analysis

Normalizing the metrics provides insight into the variability of spatial parameters across the delta (Fig. 3 shows the planview of the normalized values for all 10 metrics). Greater variability is observed in some parameters (e.g., aspect ratio, dry shape

factor, and number of outflow channels), whereas logarithmic normalization reduces spatial variability in others (e.g., island area, minimum channel width). Spatial variance in process can be seen in these maps. For example, average channel widths are smaller in the abandoned fluvial plains, but higher in the tidally-dominated portion of the delta. Similarly, the number of outflow channels draining each island is heterogeneous, with large numbers in the inactive domain of the delta and small numbers in the tidal plain.

As expected, patterns of island and channel morphology broadly match distance from major rivers or the shoreline. The largest islands are found in the central and western portions of the delta, while the smallest are rapidly changing bars and islands along major rivers. Islands in the central and western portions of the tidal zone are homogeneously small and bound by small rivers, although they form complexes that are themselves bound by larger tidal channels. Convexity captures the large scale roughness of island silhouettes. Large islands in the central and western portions of the delta, which formed by the gradual

agglomeration of islands bound by channels, have lower values of convexity. Small isolated islands in tidal regions and many of those at the active river mouth have higher values of convexity. High values of island aspect ratio follow the major fluvial pathways. Small mid-channel bars are frequently elongated, as are the bars that have accreted along the banks. Large islands forming distributary junctions in the upstream reaches of the delta also tend to have high aspect ratios, as well as some islands in the tidal region.

The results of the principal component analysis (Fig. 4) show the correlation between the normalized metrics. Dimensional metrics that describe island area show a lesser correlation to factors that scale with area (e.g., dry shape factor, number of outflow channels per island, fractal dimension), whereas those factors are strongly correlated with each other. These size metrics are inversely correlated with features that represent the shape of islands (convexity), indicating that island morphology varies systematically with island area, particularly for larger islands. The parameters that are most independent from island

area represent the characteristics of channels that bound each island (minimum, maximum, and average channel width) or the roughness of the interface between channels and islands (solidity, aspect ratio).

### 4.2   Spatial variability in process: cluster analysis

From these metric relationships, the ensemble model identified six dominant groups of islands in the GBMD according to their morphology and connectivity. Island group names were selected based on previous work classifying the delta and field-

based knowledge about the processes across the system: 'estuarine' (delta-front), 'tidal', 'fluvial', 'inactive' (upper delta plain), 'transitional' and 'other'. The results of the cluster analysis indicate spatial variability in process (Fig. 5). The patterns of the clusters fit within the three groupings presented in previous studies: there is a cluster of similar islands in the upper west (gray, 'inactive' as in Passalacqua et al., 2013), a cluster in the eastern delta plain (pinks, 'transitional' or 'active' in Passalacqua





et al., 2013) and tidal plains to the south along the Bay of Bengal (blues and purples, 'tidal' and 'estuarine', referred to as 'tidal' in Passalacqua et al., 2013).

Beyond the three physiographic regions presented in Passalacqua et al. (2013), other island clusters are identified using this method. Along the upper Ganges River, the bars formed by fluvial processes create one cluster (orange, 'fluvial'), while below the junction with the Brahmaputra, metrics are slightly different and create a separate cluster (yellow, 'fluvial'). The Meghna Estuary, where the high riverine discharges interact with the high tidal range ($\sim$4 m), also stands out with its own clusters (blues, 'tidal' and pinks, 'estuarine'). Finally, two large islands in the West Bengal region of India combine together to form a unique class (green, 'other').

These spatial groupings (Fig. 5) were clustered based on a quantitative evaluation of their 'sameness' relative to their neighbors. Each main cluster is made up of smaller, initial clusters, that share similar parameter values (Fig. 6). By comparing cluster mean values to the delta-wide mean values, patterns emerge. For example, in the 'estuarine' classes, median values for minimum, average, and maximum channel widths are much greater than the delta-wide median. The opposite is true in the clusters making up the 'transitional' class, where channel widths are low relative to the delta-wide values. Island area is also a useful parameter to explore when looking at similarities within clusters. 'Inactive' cluster values are above average, with large island complexes, whereas 'fluvial' clusters are made up of small, and sometimes transient, bars and islands.

The spread of data within each cluster and the relationship to the delta-wide metrics also show that some groups have well-constrained parameters (e.g., convexity in the 'estuarine' clusters), whereas others are more variable (e.g., number of outflow channels in the 'inactive' cluster) (Fig. 8). The similarity observed across each of the six dominant groupings provides evidence that the methodology used creates quantitatively-similar clusters.

## 4.3 Feature importance across clusters

Based on the set of metrics analyzed, we were able to identify distinct morphological classes within the GBMD. In order to understand which analyzed metric is the most indicative of the underlying process, we use the Kullback-Leibler (KL) divergence measure (Kullback and Leibler, 1951).

We compute the KL divergence between the probability density function (PDF) of each delta metric in a particular cluster and those of all other clusters in the delta. The KL divergence measures the importance of that particular delta metric for identifying islands in that cluster; divergence values greater than 1 indicate that the PDF of a delta metric for a given cluster differentiates it from the other clusters in the delta, while values less than 1 suggest that the PDF of a delta metric is similar to the PDF for other clusters in the GBMD.

We find that channel width differentiates the 'estuarine' class from the rest of the delta and dry shape factor differentiates the 'inactive' island complexes (Fig. 9). The 'other' class provides the most interesting results for feature importance, where most metrics are significantly different than the rest of the population, although quantifying the importance of metrics in this class is challenged by the small number of islands that belong to it. Differentiating between islands in the fluvial, tidal, and transitional classes from the rest of the population is more difficult, but island area shows KL divergence values right above 1 for most of these classes. Islands in the fluvial-dominated corridors are smallest, followed by the tidally controlled region,





whereas the islands in the transitional and inactive upper delta plain are distinctly larger. Dry shape factor, number of outflow channel, and convexity also contribute to differentiating some of these classes from the rest of the delta, reflecting an increasing amalgamation over the delta evolution (Fig. 9).

## 5 Discussion

### 5.1 Island clusters

#### 5.1.1 Estuarine (delta-front) class

Estuarine 1,2, and 3 are similar and consist of the islands and mouth bars at the outlet of the Ganges and Brahmaputra Rivers. These islands are characterized by a wide range of island sizes and channel widths. Most of the islands in these classes have poorly developed internal drainages and few outlet channels and are characterized by high values of solidity and above average values of convexity. Below average values of dry shape factor and of fractal dimension indicate that the outline of these islands is not complex. Aspect ratio tends to be above average, indicating that the islands are elongated.

Estuarine 4 is also composed of the islands at the mouth of rivers along the Bay of Bengal. On average, these islands are larger than other active delta-front islands. This class specifically has low minimum channel widths but high maximum and average channel widths because many of these islands are part of larger amalgamated landmasses between tidal channels that are bisected by small channels. These islands often have complex internal drainage networks and a large numbers of small outlet channels. The values of solidity are close to the overall average, while convexity values are usually below average. Higher values of dry shape factor and fractal dimension than other delta-front islands are due to the high sinuosity of the small channels bisecting the island complexes. The values of aspect ratio tend to be below average indicating these islands have a more 'stubby' shape. In a report of the Bengal Survey of 1915, these island were already characterized as unusually 'blunt-faced', due to strong ocean influence (Hirst, 1916), and perhaps wave reworking of their ocean-facing edges.

#### 5.1.2 Tidal class

Islands in the tidal class are located inland with respect to those in the estuarine class. Tidal islands in the western portion of the delta are classified as Tidal 1, while those in the central portion of the delta are classified as Tidal 2. Tidal 1 islands are larger than average and their outlines are complex, with high dry shape factors and fractal dimensions, but below average values of solidity and convexity. Islands in the Tidal 1 class also have well developed internal drainage networks and large numbers of outlet channels. Tidal 2 islands are generally smaller and show a range of morphological characteristics around the average for the delta. The two groups of islands classified as tidal are the most similar classes of islands within the delta.

#### 5.1.3 Fluvial class

Islands in the fluvial class are the smallest of the delta. The most upstream fluvial islands are part of subclass Fluvial 2, characterized by small areas and high aspect ratios, an elongated shape attributed to the dominance of unidirectional flow.





These islands show above average values of solidity and convexity and low values of fractal dimension, indicating that their outlines are simple and the channels around them are not highly sinuous. The minimum width of the channels that bound these islands is above average while the maximum and average channel widths are low, suggesting that most channels in this region are of similar size.

Islands in the Fluvial 1 class are found at the confluence of the Ganges with the Brahmaputra to downstream of the Meghna river. These small to medium islands are highly elongated and have average outline complexities. The channels in this region are uniformly wide. Most of these islands are mid channel bars or islands formed by channel cutoffs of distributary channels branching south from the Ganges River between the junctions with the Brahmaputra and Meghna rivers.

### 5.1.4   Transitional class

Islands in the Transitional 1 class are located within the known backwater zone, i.e. the upstream zone in which river flow is affected by hydrodynamic processes of the Bengal Basin. These islands are large, have many outlet channels and high dry shape factors. Channels in this zone are generally narrow and sinuous with above average fractal dimension. These islands have very low values of convexity and average to below average values of solidity. Islands in the Transitional 2 class are scattered diagonally between the estuary of the Hoogly river to the west and the junction with the Brahmaputra river to the

east, intermixing with other island classes. These islands show varied morphology but are uniformly bound by narrow channels. These two subclasses of islands are most similar to one another and together they are most similar to islands in the Tidal class.

### 5.1.5   Inactive (upper delta plain) class

The inactive upper delta plain region contains the largest islands in the delta. These islands have well developed internal drainage networks with a high number of outlet channels. The fractal dimension of their outlines is high, reflecting the high

sinuosity of the channels bounding islands. Their dry shape factors are also large, indicating that the island outlines are complex. This region spans the northern half of the delta bounded by the Hoogly and Ganges rivers. While many channels in this region are narrow, the channels that bound the islands of this class are of average width. These islands are also characterized by very low values of solidity and convexity due to their irregular shapes, while the aspect ratio varies across the group.

### 5.1.6   Other class

Two large islands in the West Bengal region of India form the class 'Other'. These are morphologically similar to islands in the inactive upper delta plain region but show lower values of solidity and higher channel widths. When channel building is restricted because of impoundment and reduced mobility of channels, large nearest edge distances and island sizes are usually observed (Syvitski and Saito, 2007; Edmonds et al., 2011).



## 5.2 Metrics importance and applicability of the approach to other systems

The classification presented in this work does not substantially differ from previous work (Passalacqua et al., 2013) and known zonations of this area (Alam, 1996) in terms of main behaviors, but a higher level of detail emerges from the analysis here proposed. This additional information allows us to capture spatial differences even among islands subject to similar processes
(e.g., the tidal and backwater zone transitional) and to characterize the full probability distribution functions of delta morphological metrics for each class. This information can be helpful for validating numerical modeling results (Angamuthu et al., 2018) in terms of correspondence between the morphology of simulated deltas and real ones.

Our classification distinguishes between islands experiencing a full spectrum of fluvial - tidal energy. The islands throughout the main corridors of the Ganges and the combined Ganges-Brahamaputra rivers stand out as having unique geometric
characteristics. Interestingly, the islands upstream and downstream of the main river confluence are slightly different morphometrically, suggesting that the relative proportion of bedload material (higher in the Brahmaputra river) or differences in bulk grainsize could play a role in setting the bar and island shapes. Along the full gradient of tidal energy impact, islands fall in different classes; tidal islands represent the dry season tidal flow extent, islands in the transitional backwater zone and islands in the inactive upper delta plain that experience no tides at all, all are morphometrically distinct. Islands at the delta front are
morphometrically unique too, partly due to the lack of internal drainage, partly due to their bounding wide channels (the classic tidal funnel shape). And perhaps because wave and tidal currents rework the immediate coastline into a blunt, stubby island shape.

Additionally our analysis provides information on which metric is most helpful at characterizing a given process. Channel width, island area, and dry shape factor are identified as important metrics across most of the clusters, indicating how the island
boundary and the complexity of island shape can be related to processes. Notably, the main fluvial corridors of these large rivers have smaller islands than the more inactive upper delta plain, testifying to the process of amalgamation over the evolution and progradation of a delta system. This same trend is apparent for the tidally-dominated zone as well, where nearshore tidal islands are (still) smaller and the more inland tidal zone has larger agglomerates.

Perhaps the most surprising result of our work is the lack of a distinct signature of human intervention on the computed delta
metrics. The anthropogenic modifications in the polder zone of the GBMD are known to have amplified tides and prevented sedimentation from previous work (Pethick and Orford, 2013; Auerbach et al., 2015), yet these modifications are not detectable in our analysis as the polders are not identified as a separate class. This result can be due to a variety of factors: first off, the image resolution may be too coarse to detect human modifications, which could act at subgrid scale with respect to the Landsat imagery used here. Also, we computed the delta metrics on the features as extracted from the imagery; the embankments are
not visible and the island boundary and properties as extracted may appear more natural than they actually are. Other metrics such as the number of outlet channels have been affected by human modifications in a visible way but have not yet modified the PDF of the metric such that it is distinguishable from the PDF of the natural islands. The formation of kash land and siltation of channels in the inland tidal zone (Wilson et al., 2017) is thought to be related to poldering and thus human-induced



modification of the tidal prism, but our cluster analysis also shows how infill of the channel network and amalgamation of young islands over time is an ongoing morphological change with maturation of the delta plain.

The approach here proposed is applicable to any system, provided that the island and channel sample is large enough to yield robust statistics. Because of the difficulty in extracting delta networks and the manual labor involved, studies up to date

have analyzed metrics only in small systems or have focused on bulk metrics that do not capture how the characteristics of the delta may vary spatially and temporally. We expect that further development in automatic approaches for delta network extraction (Isikdogan et al., 2017b, 2018) and for the analysis of network change over time (Jarriel et al., 2019) will enable similar analyses at the global scale and over time.

## 6  Conclusions

In this work we presented a machine learning approach for the analysis of river deltas based on remotely sensed imagery. The approach relies on a set of delta metrics and their statistical distributions, to identify similarities among clusters of islands. The approach identifies six major zones within the delta that can be related to the processes acting on the system. The method does not distinguish between polders and natural islands, suggesting that at the resolution of Landsat imagery, human modifications have not yet left an imprint on island morphology. The approach here proposed is applicable to any delta with a large enough

number of islands to compute statistical distributions and provides information relevant to the validation of numerical models and to understanding which delta metrics carry the most information on a given process.

*Code and data availability.*  The code and data are available at https://github.com/csdms-contrib/DeltaClassification and also available via CSDMS repository https://csdms.colorado.edu/wiki/Model:DeltaClassification.

*Acknowledgements.*  This material is based on work supported by the National Science Foundation grant number OCE-1600222 (to PP and

IO), and EAR-1719670 and CAREER-1350336 awarded to PP. Funds donated by S. Ullah to support MP are also gratefully acknowledged. The authors would also like to thank the network of researchers affiliated with the NSF funded Coastal SEES project on the Ganges-Brahmaputra-Meghna Delta for useful and interesting discussions on the results presented here.



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



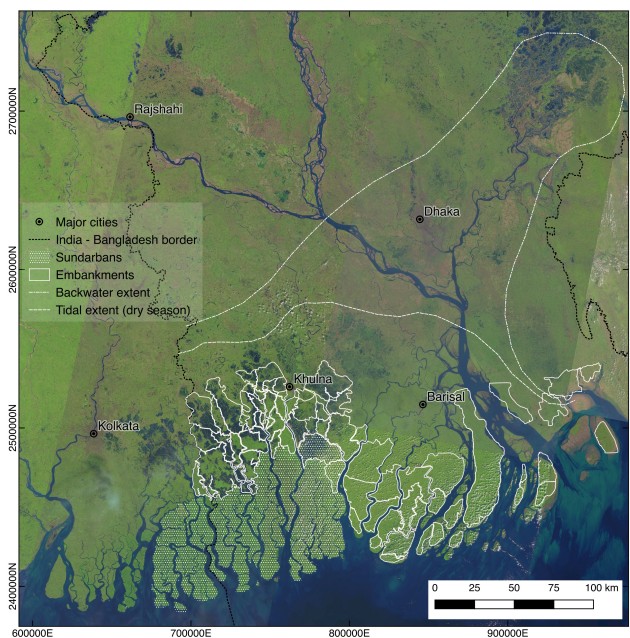

**Figure 1.** Natural color Landsat 8 image of the GBMD from 2018. Indicated on this map are polders (white solid lines), Sundarbans (white dot grid), approximate dry season tidal limit (white dash line) and the dry season backwater extent (white dash-dot line).



| PARAMETER | DEFINITION | LOW | HIGH |
|---|---|---|---|
| Area | | | |
| Aspect Ratio | Major axis/ Minor axis | | |
| Dry Shape Factor | Dry Perimeter/ sqrt(Area) | | |
| Fractal Dimension | log(#boxes)/ log(magnification) | | |
| Solidity | Area/ Area of Convex Hull | | |
| Channel Width | Min, Max, Average | | |
| Number of Outflow Channels | | | |
| Convexity | Perim. Convex Hull/Dry Perim. | | |

**Figure 2.** Illustration of the metrics used in this analysis. As we use max, average, and min for channel width, the total number of metrics is 10.





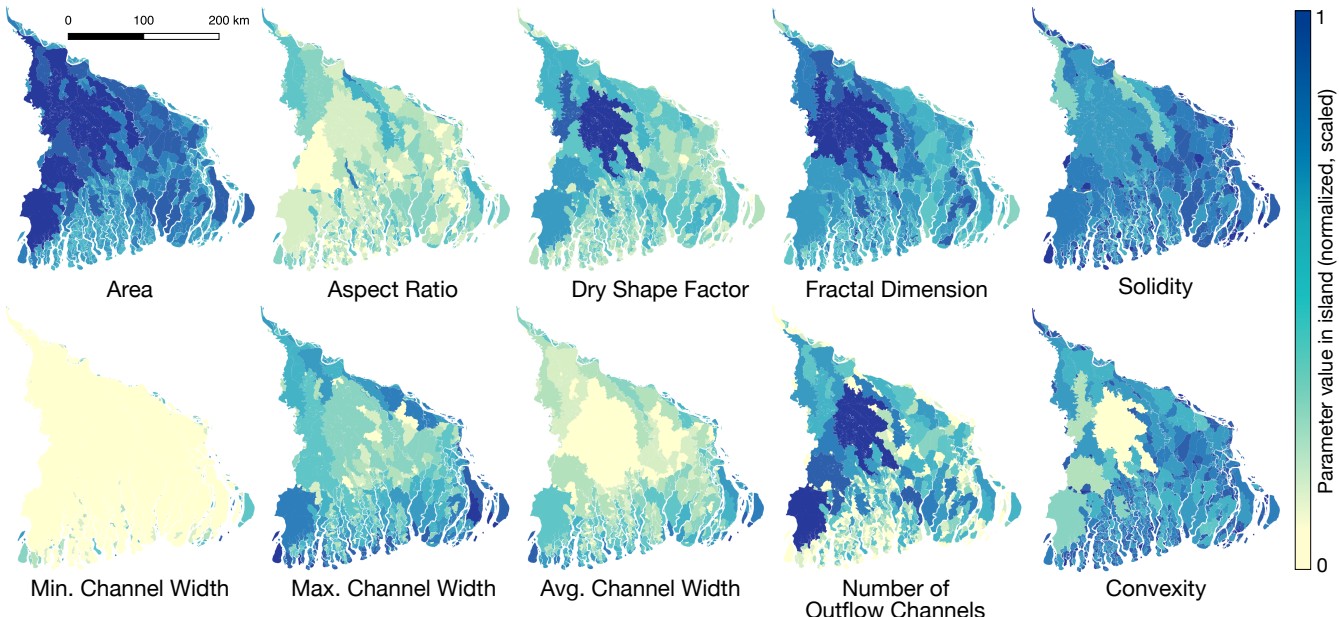

**Figure 3.** Map view of metrics, normalized, and scaled. The metrics are normalized using a logarithmic normalization and scaled between 0 and 1.





**Figure 4.** Variable correlation matrix. The metrics used in this analysis show moderate degrees of correlation.





**Figure 5.** Delta classification. The islands are classified into 14 clusters belonging to 6 main classes.



**Figure 6.** Dendrogram of classification and heatmap of variables. Warm colors indicate a high median value of a given parameter relative to the delta-wide median, while cool values indicate a low cluster median relative to the delta-wide value.



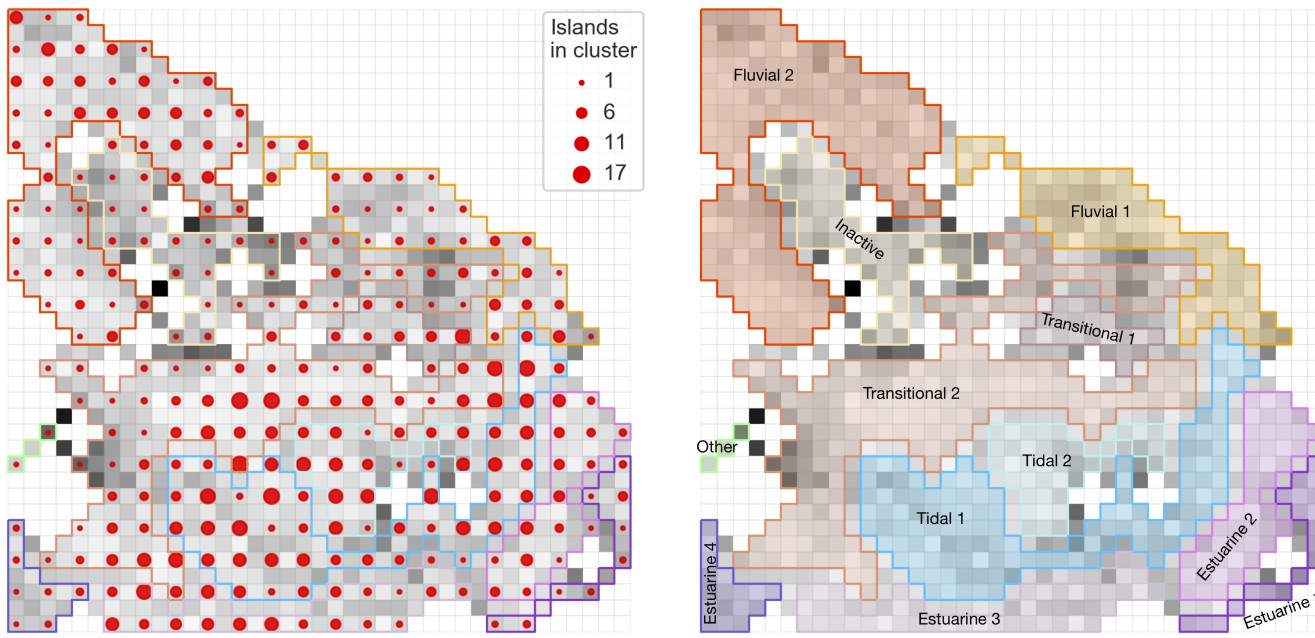

**Figure 7.** U-matrix. Islands are clustered into their adjacent groups.





Earth **Surface**
**Dynamics**
Discussions

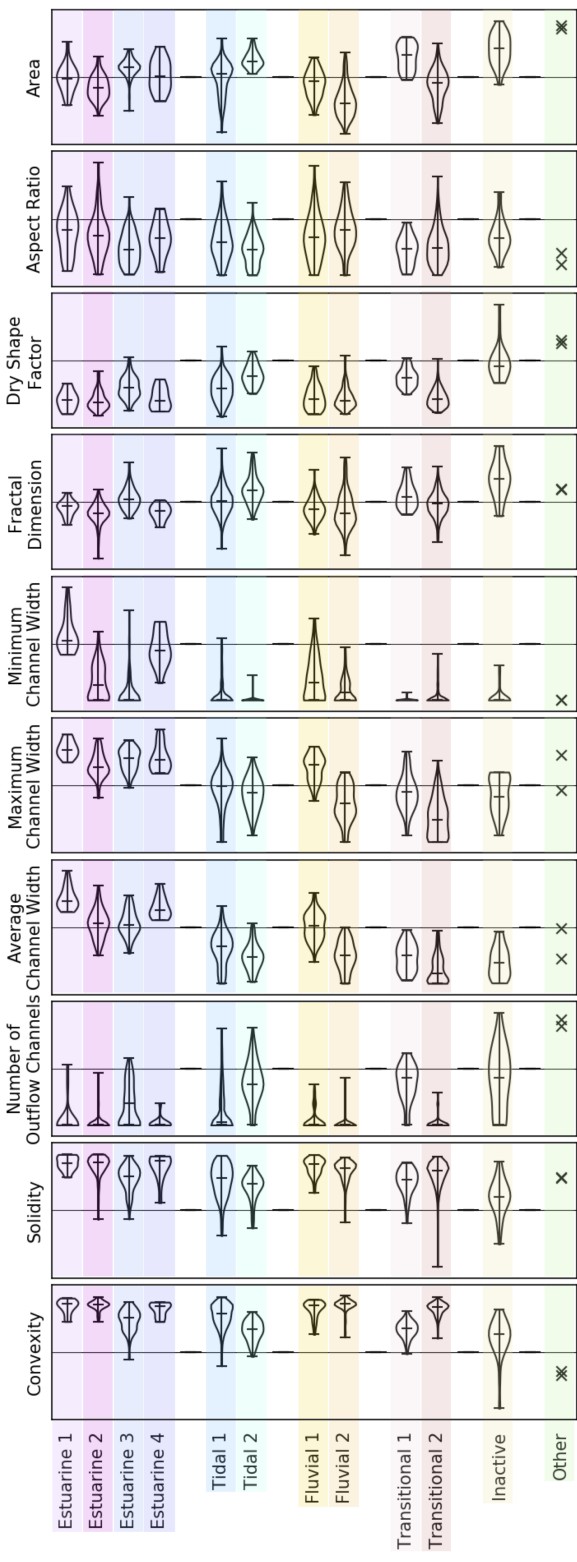

**Figure 8.** Violin plots of parameter values for each class. The analysis allows the calculation of the PDF of each metric for each cluster.

Earth **Surface**
Dynamics
Discussions



**Figure 9.** Kullback-Leibler divergence between the distribution of parameter values for islands of each class and all other islands for each variable. A value greater than 1 indicates that the variable differentiates the population of islands in that class from those in other classes. A value less than 1 shows that the distribution of values of that variable for that island class are similar to values for other islands.