# Peer review of "Dominant process zones in a mixed fluvial-tidal delta are morphologically distinct"

_Earth Surface Dynamics, 2020_

## Referee Comment (RC1) · Anonymous Referee #1 · 23 Apr 2020

This is a very interesting and thought-provoking article that I enjoyed reading. The main contribution of the paper is to set out a new method of classifying zones of homogenous morphology using a machine learning algorithm. The method is illustrated using an example from a delta, specifically the Ganges (GBM) delta. In essence, the approach works by using remote sensing imagery to identify the patterns of channels and islands in the delta, employs a range of morphological metrics, and then the ML algorithm builds and identifies a network of homogenous zones across the area of interest. The result in this study is a map that classifies the delta into 6 main morphologically discrete zones (that are themselves further sub-divided), which in principle relate to different process dynamics in each area. Although in some respects we do not learn anything completely new about the morphological zones of the GBM (a point

I return to below), the method is exciting and interesting and its potential is well illus-trated with this particular case study. Specifically, the very large extent of this delta (and its complexity) illustrates very well the potential of the approach to identify pro-cess/morphological zones of variable dimensions and across a large area. There is no doubt that the approach has potential to be employed in other morphological settings and, taken together with the point that the methodological aspects of the paper appear to be delivered robustly, I do think that the criteria for publication (significance, scientific quality, and presentation quality) are met.

Having said that, there are some aspects of the paper where I did feel that the overall significance and originality of the work could be better highlighted. Specifically, I had the sense that the paper as it stands does a very good job of describing the method, but it is not as strong at explaining the significance of the work in terms of showing clearly how the method could be applied to gain insight into morphodynamic processes. The paper as it stands builds a classification of process/morpho zones based on a mosaic of satellite images dating from 1990 – if it were possible to repeat the method, using much more recent imagery, it would presumably be possible to demonstrate more clearly how these morphological zones have evolved in space-time in response to some of the changing process drivers that the authors speculate on in their discussion. I do accept that undertaking that analysis is not a trivial task and I emphasise that the paper is acceptable as it stands, it's just that the paper could be magnificent if such an analysis were also included. Admittedly, that magnificence could equally be achieved in an additional paper somewhere down the line!

Irrespective of that suggestion, I have a number of other specific suggestions that could potentially be addressed to clarify further some aspects of the discourse. I now list those here:

1) On p3, L13, the citation to Meshkova and Carling (2013) is slightly misleading. The sentence implies that their paper is about deltas, but it is about a stretch of river well upstream of one. It just needs to be rephrased to make it clear that the process they

used could presumably be applied to deltas. 2) At the end of the introduction (the paragraph at p3-4), I think it would be helpful for the reader to have a stronger and explicit statement here concerning the overall aims and objectives of the paper, but also in particular the originality and significance of the work. How does the new method build on previous approaches and what does that mean for potentially enhancing our understanding of (delta) morphodynamics? In fact this is slightly a recurring theme through the paper. The main result (Figure 5) provides a classification that is not that dis-similar (it does have much more detail) to previous classifications (the authors recognise this as the classes are in essence taken from that prior work). So elsewhere in the paper too the extent to which the new work really offers new insight needs to be discussed and addressed (this concern partly motivates my main suggestion above). 3) In the methods section (p5, final paragraph) I felt that a little bit more detail could be provided regarding the source imagery, rather than relying exclusively on the citation to Passalacqua et al (2013). When I went and read that paper I only then realised that the imagery being used was acquired in 1990, which of course means that the classification that is developed of the GBM's morphodynamic zones is one that is pertinent to conditions three decades ago....this point at least needs to be absolutely clear to the reader. 4) In turn, this point raises some further questions about the way in which the mapping results are interpreted. Throughout the paper it is implicitly and explicitly assumed that the distinctive morphological zones represent distinctive *process* zones. This is not an unreasonable assumption, but presumably the morphological zones represent the outcome of ongoing dynamic processes whose response and relaxation times vary according to spatial scale, such that the observed morphology is presumably not instantaneously responsive of process conditions in 1990, but rather also reflect process conditions years or decades before. I would like to see a clearer discussion of this point because this would aid in understanding how a classification approach such as this – which is geared towards working at large spatial scales because of the advantages of the remote sensing and ML techniques – can adequately reflect (lagged) processes operating over those large spatial scales. The answer to

that question fundamentally conditions the utility and hence significance of the new approach in terms of understanding process dynamics. Also on this theme, I wonder if the paragraph from L24 in section 5.2 should also be modified to reflect this point more closely. Specifically, the authors comment on their analysis not yet detecting the amplified tides and prevented sedimentation identified in previous work – but in relation to the Pethick and Orford study there is only a single gauge (at Khulna) that has any data before the 1990 acquisition date of this study, and actually the increase before and up to 1990 (the period that is pertinent to the acquired imagery, as per point 3) is not very large. So, I am not surprised that they find this result, it's just that my interpretation of why the authors do not is different from theirs (image resolution, feature discrimination, etc).

Finally, I would like to suggest also that some minor modifications to some of the Figures (which in general are very good) may help the overall clarity of communication. In particular: • I found it very hard to discriminate the two white lines demarking the tidal extent and backwater extent on Figure 1. Perhaps the use of an alternative (bright) colour for one of these lines would help? • On Figure 4, it was not clear to me (presumably they mark ranges of correlation coefficients) what the significance of the coloured shading of the cells in the matric represents. Either a legend needs to be added to illustrate the meaning of the shading, or perhaps add some detailed text to the figure caption for the same? By the same token, there is not enough information in the figure caption to be able to understand what the line and scatter plots actually represent (at least without extensive cross-referencing to the text). • I wondered – this is not a critical point – if the logical sequence of diagrams should actually be that Figures 6 and 7 (which outline in effect the inputs into the overall classification) should precede Figure 5, which is the outcome of the classification process? In any case, as with Figure 4 I felt that the figure caption and legend for Figure 5 could be a little bit more detailed. It took me a little bit of work to figure out exactly what the 6 classes in the caption are (and the 14 clusters) – adding the details of the 6 classes to the caption would be helpful.

**ESurfD**

---

## Referee Comment (RC2) · Anonymous Referee #2 · 27 May 2020

The manuscript describes a data analysis workflow to classify regions of the Ganges-Brahmaputra-Meghna Delta (GBMD) based on island morphometrics. These island and channel morphometrics correspond to processes and process zones. I found the paper well written, clear, and interesting. I have mostly minor comments (listed below). Before I get to these minor, specific comments, I have one larger comment focused on the extensibility of the analysis. As stated on P4 L5-6, the GBMD is the largest subaerial delta in the world. The manuscript describes working with a dataset of 1200 features (P5 L31). I am wondering about applying this to other deltas around the world – how many deltas would work with this technique? The manuscript doesn't necessarily need to answer that question, but I think the paper would benefit from a paragraph where future use of this technique was specifically – specific ways in

which this analysis could be extended to understand other deltas on earth (which will therefore have smaller sample sizes of island/channel features). How can researchers who study other deltas use this technique? Is there a threshold sample size where it stops being useful? Could a researcher assemble a large dataset of features from many different deltas then use a modified version of the workflow to do inference on new delta islands (from random deltas) to determine if the random island is dominated by certain processes?

To be very clear – I like this work, it is interesting to me, I would just like an explicit moment where the manuscript steps back and looks at how this neat workflow could be used by others in other settings.

Specific Comments

P1 L2: Can you define 'resilient' in this context?

P1 L17-18: I think a more precise way to convey this idea is: the data, analysis techniques, and chosen morphometrics do able to detect human modifications to the system. This is said in a more defensible way in the discussion (P12 L24)

P2 L4: Can you define 'resilient' in this context?

P4 L1-2: this is a really neat idea, but i did not see this discussed in the discussion/conclusion.

P5 L15: this line is confusing because of the repetition of the words 'cluster and 'group':

"…island are clustered into groups… clusters are grouped'

Do I understand step 4 correctly that the groups are further combined and ordered?

P6 L6: I'm not sure I agree with this statement as it is written – can you provide a citation for this?

P7 L3: ClusterPy should be cited — look at the 'readme.md' on the github page for

the details: https://github.com/clusterpy/clusterpy Also – if there are any other software packages you used i would cite them (software is often not cited in text, but should be).

P6 and P7, generally: are there any subjective hyperparameters that you set? thresholds for the clustering algorithms? If so, please point those out or mention them.

P8 L 20: To me, figure 4 displays the results of the correlation analysis, not the results from the PCA.

P9 L9: I suggest removing the scare quotes on 'sameness', since the dissimilarity metric is discussed in sect. 3.4.

P12 L4: I think the word 'proposed' can be deleted.

Data and Software repository: I see this on github https://github.com/csdms-contrib/DeltaClassification. I would recommend a 'Readme.md' file to help explain the code to make it reproduceable/ extensible for future work. i.e., telling folks where to find the data (i see some '.shp' files in the '_input' folder), and how they could use the code.. I would also provide a DOI for the code itsefl (via the Zenodo integration with Github) and provide the citation for the code in this manuscript.

Figure 1: I cannot tell the difference between all of the white outlines.

---

## Author Comment (AC1) · 1 Jul 2020

Thank you very much to both reviewers for the helpful comments and feedback on our manuscript. Below our replies to each comment and an explanation of how the manuscript will be revised to address each point.

Reviewer #1

(R1.1) This is a very interesting and thought-provoking article that I enjoyed reading. The main contribution of the paper is to set out a new method of classifying zones of homogenous morphology using a machine learning algorithm. The method is illustrated using an example from a delta, specifically the Ganges (GBM) delta. In essence, the approach works by using remote sensing imagery to identify the patterns of chan-

nels and islands in the delta, employs a range of morphological metrics, and then the ML algorithm builds and identifies a network of homogenous zones across the area of interest. The result in this study is a map that classifies the delta into 6 main morphologically discrete zones (that are themselves further sub-divided), which in principle relate to different process dynamics in each area. Although in some respects we do not learn anything completely new about the morphological zones of the GBM (a point I return to below), the method is exciting and interesting and its potential is well illustrated with this particular case study. Specifically, the very large extent of this delta (and its complexity) illustrates very well the potential of the approach to identify process/morphological zones of variable dimensions and across a large area. There is no doubt that the approach has potential to be employed in other morphological settings and, taken together with the point that the methodological aspects of the paper appear to be delivered robustly, I do think that the criteria for publication (significance, scientific quality, and presentation quality) are met.

Reply: Thank you for your positive assessment of our manuscript. We agree that overall the classification resulting from the proposed method is not too different from our previous one [Passalacqua et al., 2013]. However, there is an additional level of information resulting from this method that allows us to capture at a finer 'resolution' the morphological similarities among delta islands and create groupings of 'similar' islands. One of the motivations of this work is in fact capturing the unique aspects of a complex network so that individual islands, representative of each group, can be modeled at high resolution and results scaled up to the other islands in the same group. We did not necessarily expect this method to give us different results than our previous method (as those groups are related to processes acting on the delta), but rather to obtain a more detailed classification to inform numerical modeling and further interpret existing morphological zones. Revisions will be made to state more clearly the significance and novelty of this work as explained in our replies below to the reviewer's specific comments.

(R1.2) Having said that, there are some aspects of the paper where I did feel that the overall significance and originality of the work could be better highlighted. Specifically, I had the sense that the paper as it stands does a very good job of describing the method, but it is not as strong at explaining the significance of the work in terms of showing clearly how the method could be applied to gain insight into morphodynamic processes.

Reply: Thank you for your comment; we have attempted to highlight how the method provides new insight in the Discussion section where we interpret the geomorphological meaning of each class. However, your comment clearly points us to the need to reinforce this aspect of the paper. As also mentioned in other replies below, we will add several statements in the revised manuscript: at the end of the introduction "In this paper, we propose a two-step unsupervised machine learning method to analyze spatial patterns in large river deltas. We apply this method to the Ganges-Brahmaputra-Meghna Delta and show that by clustering areas of the distributary system with common morphological characteristics, we are able to reliably extrapolate our understanding of the physical processes that locally dominate island and channel morphology across large areas of the delta. While our previous work highlighted three main regions in the GBMD delta (Passalacqua et al., 2013), here we are able to extract more information on the delta surface network, which can be used to increase the explanatory power of on-the-ground observations and guide future field work and the selection of representative islands for high-resolution numerical modeling (and even for coastal zone management practices). Additionally, we provide a method for identifying which metrics are most useful in differentiating process signatures, thus providing guidance on what properties to measure in other systems."

(R1.3) The paper as it stands builds a classification of process/morpho zones based on a mosaic of satellite images dating from 1990 – if it were possible to repeat the method, using much more recent imagery, it would presumably be possible to demonstrate more clearly how these morphological zones have evolved in space-time in response to

some of the changing process drivers that the authors speculate on in their discussion. I do accept that undertaking that analysis is not a trivial task and I emphasise that the paper is acceptable as it stands, it's just that the paper could be magnificent if such an analysis were also included. Admittedly, that magnificence could equally be achieved in an additional paper somewhere down the line!

Reply: Thank you for your comment; indeed you raise an important point. The imagery used in this paper is in fact the same satellite imagery used in our previous paper, as we could rely on the previously extracted network for comparison. We have performed the analysis that you suggest in another manuscript: we have developed a method for quantifying the variance of the network through time and validated it on experimental results [Jarriel et al., 2019] and then applied the method on LANDSAT imagery for the period 1989-2019 [Jarriel et al., in revision]. There are certainly changes detected, particularly along the main rivers and in the polder zone. We will cite this work in our revised manuscript and provide a brief discussion on how the analysis proposed in this paper can be applied through time to quantify whether local changes observed in the network are reflected in the machine learning classification.

Jarriel, T., F. Isikdogan, A. Bovik, P. Passalacqua (2019), Characterization of deltaic channel morphodynamics from imagery time series using the Channelized Response Variance, Journal of Geophysical Research - Earth Surface, 124, 3022-3042, doi:10.1029/2019JF005118.

Jarriel, T., L. F. Isikdogan, A. Bovik, P. Passalacqua, System wide channel network analysis reveals hot-spots of morphological change in anthropogenically modified regions of the Ganges Brahmaputra Meghna Delta, Scientific Reports, in revision.

This section of the Discussion will be edited as follows: "Perhaps the most surprising result of our work is the lack of a distinct signature of human intervention on the computed delta metrics. The anthropogenic modifications in the polder zone of the GBMD are known to have amplified tides and prevented floodplain sedimentation (Pethick and

Orford, 2013; Auerbach et al., 2015), yet these modifications are not detectable in our analysis as the machine learning techniques do not identify the polders as a separate class. This result can be due to a variety of factors: first off, the mosaic used as input imagery is from the 1990s and while polders at that point had been in place for three decades (since the 1960's), their signature may not be visible yet. Additionally, the resolution may be too coarse to detect human modifications, which could act at subgrid scale with respect to the Landsat imagery used here. We computed the delta metrics on the features as extracted from the imagery; the embankments are not visible and the island boundary and properties as extracted may appear more natural than they actually are. Furthermore, embankments are usually built to follow the natural edges and contours of the islands, in a way, embankments 'freeze' island geometry in place. Other metrics such as the number of outlet channels have been affected by human modifications in a visible way but have not yet modified the PDF of the metric such that it is distinguishable from the PDF of the natural islands. The formation of new 'kash' land and siltation of channels in the inland tidal zone (Wilson et al., 2017) is thought to be related to poldering and thus human-induced modification of the tidal prism, but our cluster analysis also shows how infill of the channel network and amalgamation of young islands over time is an ongoing morphological change with maturation of the delta plain. Repeating this analysis on time series imagery of the GBMD with tools capable of quantifying change (Jarriel et al., 2019) provides additional information and points to the polder region as an area of change over the last three decades (Jarriel et al., in revision). These changes may have not impacted yet the overall classification presented in this work; the question of what disturbance size affects the system as a whole is an important one which is yet to be addressed."

(R1.4) Irrespective of that suggestion, I have a number of other specific suggestions that could potentially be addressed to clarify further some aspects of the discourse. I now list those here: 1) On p3, L13, the citation to Meshkova and Carling (2013) is slightly misleading. The sentence implies that their paper is about deltas, but it is about a stretch of river well upstream of one. It just needs to be rephrased to make it

clear that the process they used could presumably be applied to deltas.

Reply: Thank you for catching that - we will revise the text to better represent the goals of that work as follows: "Spatial variability in the morphology of the channel network also results in diversity in the geometry of islands in multi-threaded systems (Meshkova and Carling, 2013), which in deltas have been quantified with multiple metrics of island and network morphology (Edmonds et al., 2011), with statistical analyses (Passalacqua et al., 2013), and in numerical models under various input conditions, sea level rise, and subsidence (Liang et al., 2016b, a)."

(R1.5) 2) At the end of the introduction (the paragraph at p3-4), I think it would be helpful for the reader to have a stronger and explicit statement here concerning the overall aims and objectives of the paper, but also in particular the originality and significance of the work. How does the new method build on previous approaches and what does that mean for potentially enhancing our understanding of (delta) morphodynamics? In fact this is slightly a recurring theme through the paper. The main result (Figure 5) provides a classification that is not that dis-similar (it does have much more detail) to previous classifications (the authors recognise this as the classes are in essence taken from that prior work). So elsewhere in the paper too the extent to which the new work really offers new insight needs to be discussed and addressed (this concern partly motivates my main suggestion above).

Reply: Thank you, we agree with you and we will add the following paragraph at the end of the introduction: "In this paper, we propose a two-step unsupervised machine learning method to analyze spatial patterns in large river deltas. We apply this method to the Ganges-Brahmaputra-Meghna Delta and show that by clustering areas of the distributary system with common morphological characteristics, we are able to reliably extrapolate our understanding of the physical processes that locally dominate island and channel morphology across large areas of the delta. While our previous work highlighted three main regions in the GBMD delta (Passalacqua et al., 2013), here we are able to extract more information on the delta surface network, which can be used

to increase the explanatory power of on-the-ground observations and guide future field work and the selection of representative islands for high-resolution numerical modeling (and even for coastal zone management practices). Additionally, we provide a method for identifying which metrics are most useful in differentiating process signatures, thus providing guidance on what properties to measure in other systems." And we will also add this statement to the Discussion: "Which metrics are most effective at capturing the signature of geomorphic processes is an open question (Edmonds et al., 2011; Liang et al., 2016); our analysis results provide guidance on what to measure and the relative importance of these metrics in a delta as large and heterogeneous as the GBMD."

(R1.6) 3) In the methods section (p5, final paragraph) I felt that a little bit more detail could be provided regarding the source imagery, rather than relying exclusively on the citation to Passalacqua et al (2013). When I went and read that paper I only then realised that the imagery being used was acquired in 1990, which of course means that the classification that is developed of the GBM's morphodynamic zones is one that is pertinent to conditions three decades ago. . ..this point at least needs to be absolutely clear to the reader.

Reply: Thank you for the comment and as stated above, you are absolutely correct. We will provide more details on the data source as follows: "We obtained the channel network and water surface mask from Orthorectified Landsat Thematic Mapper Mosaics (Landsat GeoCover TM 1990 Edition Mosaics, tiles N–45–20 and N–46–20, 28.5 m resolution) as in Passalacqua et al., (2013)."

(R1.7) 4) In turn, this point raises some further questions about the way in which the mapping results are interpreted. Throughout the paper it is implicitly and explicitly assumed that the distinctive morphological zones represent distinctive *process* zones. This is not an unreasonable assumption, but presumably the morphological zones represent the outcome of ongoing dynamic processes whose response and relaxation times vary according to spatial scale, such that the observed morphology is presumably not instantaneously responsive of process conditions in 1990, but rather

also reflect process conditions years or decades before. I would like to see a clearer discussion of this point because this would aid in understanding how a classification approach such as this – which is geared towards working at large spatial scales because of the advantages of the remote sensing and ML techniques – can adequately reflect (lagged) processes operating over those large spatial scales. The answer to that question fundamentally conditions the utility and hence significance of the new approach in terms of understanding process dynamics.

Reply: Thank you for raising such an important point; definitely, the morphology observed and detected at any given time, although the input data are in fact an instantaneous image of the system, does not represent only instantaneous conditions (e.g., discharge and thus river widths), but also characteristics of the system resulting from processes acting at long time scales. So effectively the imagery captures both instantaneous conditions as well as the long term signature of physical processes. We believe that classifications such as that proposed here, or in our previous work, capture predominantly the morphology resulting from long term processes: large-scale delta plain progradation that defines the tidal impact and infill of the channel network, and main channel belt avulsions. While instantaneous conditions can modify the network, those variations would be observed in the monsoon season (when satellite imagery is cloudy) in terms of channel width, but statistically the network would remain the same. It is indeed a very interesting research question as to what scale a local disturbance is felt at the network scale and it is a question we are currently working on. We will revise this part of the Discussion as follows (see also reply to R1.3): The anthropogenic modifications in the polder zone of the GBMD are known to have amplified tides and prevented floodplain sedimentation (Pethick and Orford, 2013; Auerbach et al., 2015), yet these modifications are not detectable in our analysis as the ML techniques do not identify the polders are not identified as a separate class. This result can be due to a variety of factors: first off, the mosaic used as input imagery is from the 1990s and while polders at that point had been in place for three decades (since the 1960's), their signature may not be visible yet. Additionally, the resolution may be too coarse to detect human modifications, which could act at subgrid scale with respect to the Landsat imagery used here. We computed the delta metrics on the features as extracted from the imagery; the embankments are not visible and the island boundary and properties as extracted may appear more natural than they actually are. Furthermore, embankments are usually built to follow the natural edges and contours of the islands, in a way, embankments 'freeze' island geometry in place.thus reducing their impact on island geometry. Other metrics such as the number of outlet channels have been affected by human modifications in a visible way but have not yet modified the PDF of the metric such that it is distinguishable from the PDF of the natural islands. The formation of new 'kash' land and siltation of channels in the inland tidal zone (Wilson et al., 2017) is thought to be related to poldering and thus human-induced modification of the tidal prism, but our cluster analysis also shows how infill of the channel network and amalgamation of young islands over time is an ongoing morphological change with maturation of the delta plain. Repeating this analysis on time series imagery of the GBMD with tools capable of quantifying change (Jarriel et al., 2019) provides additional information and points to the polder region as an area of change over the last three decades (Jarriel et al., in revision). These changes may have not impacted yet the overall classification presented in this work; the question of what disturbance size affects the system as a whole is an important one which is yet to be addressed."

(R1.8) Also on this theme, I wonder if the paragraph from L24 in section 5.2 should also be modified to reflect this point more closely. Specifically, the authors comment on their analysis not yet detecting the amplified tides and prevented sedimentation identified in previous work – but in relation to the Pethick and Orford study there is only a single gauge (at Khulna) that has any data before the 1990 acquisition date of this study, and actually the increase before and up to 1990 (the period that is pertinent to the acquired imagery, as per point 3) is not very large. So, I am not surprised that they find this result, it's just that my interpretation of why the authors do not is different from theirs (image resolution, feature discrimination, etc).

Reply: Thank you for this comment and you are absolutely correct. As the imagery is from the 1990s it is possible that anthropogenic modifications were not yet visible in the system. Our work in Jarriel et al. [in revision] does suggest large changes in the polders. The extent to which these changes will impact the network classification, given the image resolution, needs to be quantified in future work. We will revise the abstract as follows: "The method is not able to distinguish between islands with embankments (polders) and natural islands in the nearby mangrove forest (Sundarbans), suggesting that human modifications have not yet altered the gross geometry of the islands beyond their previous 'natural' morphology or that the input data (time, resolution) used in this study are preventing the identification of a human signature." and the Discussion as mentioned in previous replies (R1.3 and R1.7).

(R1.9) Finally, I would like to suggest also that some minor modifications to some of the Figures (which in general are very good) may help the overall clarity of communication. In particular: I found it very hard to discriminate the two white lines demarking ′ the tidal extent and backwater extent on Figure 1. Perhaps the use of an alternative (bright) colour for one of these lines would help?

Reply: Thank you, we will revise the figure. This point was also raised by Reviewer 2.

(R1.10) On Figure 4, it was not clear to me (presumably they mark ranges of correlation coefficients) what the significance of the coloured shading of the cells in the matrix represents. Either a legend needs to be added to illustrate the meaning of the shading, or perhaps add some detailed text to the figure caption for the same? By the same token, there is not enough information in the figure caption to be able to understand what the line and scatter plots actually represent (at least without extensive cross-referencing to the text).

Reply: We agree and we will add the following explanatory text to the Figure 4 caption: "Matrix of correlation values for sets of metrics. Warm colors indicate pairs of metrics that are negatively correlated. Darker warm colors indicate stronger negative

correlation relationships (e.g. the convexity-dry shape factor element). Cool colors indicate pairs of metrics that are positively correlated, with darker cool colors indicating stronger positive correlations (e.g. the average channel width-maximum channel width element). Overall, metrics used in this analysis show moderate degrees of correlation. Also shown are the distributions for each metric value (e.g. the area-area element) as well as the scatter plot distribution of the metric relationships (e.g. the area-aspect ratio element)."

(R1.11) I wondered – ′ this is not a critical point – if the logical sequence of diagrams should actually be that Figures 6 and 7 (which outline in effect the inputs into the overall classification) should precede Figure 5, which is the outcome of the classification process?

Reply: Thanks for the comment; we think the current order is correct as information from Figure 5 is used in the following ones. To aid the reader in following the workflow, we have revised the corresponding figure captions. Edits to the caption of Figure 5 are addressed in the next comment.

Figure 6: "Dendrogram of classification and heatmap of variables. Warm colors indicate a high median value of a given parameter relative to the delta-wide median, whereas cool values indicate a low cluster median relative to the delta-wide median value."

Figure 7: "The U-matrix (unified distance matrix) visualizes the number of adjacent islands within a node. Larger dots represent a greater number of islands, as determined by the GeoSOM method. Smaller dots represent a smaller number of islands. The colored outlines and shaded areas correspond to the 6 main classes or groups."

Additionally, the following text was added to the manuscript section that discusses Figure 7: "The U-matrix, or unified distance matrix, shows the result of the GeoSOM analysis, illustrates the number of adjacent islands assigned to each node within a group (Fig. 7)."

(R1.12) In any case, as with Figure 4 I felt that the figure caption and legend for Figure 5 could be a little bit more detailed. It took me a little bit of work to figure out exactly what the 6 classes in the caption are (and the 14 clusters) – adding the details of the 6 classes to the caption would be helpful.

Reply: We agree and we will add the following text in the figure caption:

Figure 5: "Island classification as a result of the hierarchical agglomerative clustering method, using a geographic constraint so that only adjacent islands can be grouped. Each island in the GBM delta is classified into 12 individual clusters. The 12 individual clusters are further grouped into 6 main classes, using a dendrogram (Fig. 6). The 6 main groups include estuarine (purples), tidal (blues), transitional (pinks), inactive (gray), fluvial (oranges) and other (green)."

Reviewer #2

(R2.1) The manuscript describes a data analysis workflow to classify regions of the Ganges-Brahmaputra-Meghna Delta (GBMD) based on island morphometrics. These island and channel morphometrics correspond to processes and process zones. I found the paper well written, clear, and interesting.

Reply: Thank you for your positive assessment of our manuscript.

(R2.2) I have mostly minor comments (listed below). Before I get to these minor, specific comments, I have one larger comment focused on the extensibility of the analysis. As stated on P4 L5-6, the GBMD is the largest subaerial delta in the world. The manuscript describes working with a dataset of 1200 features (P5 L31). I am wondering about applying this to other deltas around the world – how many deltas would work with this technique? The manuscript doesn't necessarily need to answer that question, but I think the paper would benefit from a paragraph where future use of this technique was specifically – specific ways in which this analysis could be extended to understand other deltas on earth (which will therefore have smaller sample sizes of island/channel

features). How can researchers who study other deltas use this technique? Is there a threshold sample size where it stops being useful? Could a researcher assemble a large dataset of features from many different deltas then use a modified version of the workflow to do inference on new delta islands (from random deltas) to determine if the random island is dominated by certain processes? To be very clear – I like this work, it is interesting to me, I would just like an explicit moment where the manuscript steps back and looks at how this neat workflow could be used by others in other settings.

Reply: Thank you for raising this important point, which is not easy to answer without a scaling analysis, which would be outside the scope of this work, and such analysis for other delta systems would increase the complexity of the paper. It's an important point to bring up though and we will add the following paragraph to the Discussion: "The approach here proposed would be applicable to any system, provided that the island and channel sample is large enough to yield robust statistics and the application of a machine learning approach. The actual number of islands needed will also depend on the strength of the geomorphic signature (signal) versus the delta's heterogeneity (noise). This signal to noise ratio may also influence the applicability of our method to the classification of islands from many deltas to identify similarities and process signatures across systems, rather than within one system only as in the analysis we performed."

(R2.3) Specific Comments P1 L2: Can you define 'resilient' in this context?

Reply: Thank you, we have added the definition later in the introduction as per your point about P2 L4.

(R2.4) P1 L17-18: I think a more precise way to convey this idea is: the data, analysis techniques, and chosen morphometrics do able to detect human modifications to the system. This is said in a more defensible way in the discussion (P12 L24)

Reply: Thank you, we will revise the text as follows: "The method is not able to distinguish between islands with embankments (polders) and natural islands in the nearby
mangrove forest (Sundarbans), suggesting that human modifications have not yet altered the gross geometry of the islands beyond their previous 'natural' morphology. However, the lack of a human signature may be due to the input data (time, resolution) used in this study."

(R2.5) P2 L4: Can you define 'resilient' in this context?

Reply: Thank you for this comment, we use resilient as in Hoitink et al. [2020], a system that is capable of recovering from extreme events and of sustaining itself. We will add the following sentence: ". As in Hoitink et al. (2020), we define resilient a system that is capable of recovering from extreme events and of sustaining itself."

(R2.6) P4 L1-2: this is a really neat idea, but i did not see this discussed in the discussion/conclusion.

Reply: Thank you, that is a good point and we have decided to move this statement to the Discussion. The text will be placed at the end of the Discussion, revised as follows: "Hierarchical clustering of delta islands according to their common characteristics can also allow the identification of areas of the landscape that would be affected by different forecasted scenarios of future environmental conditions."

(R2.7) P5 L15: this line is confusing because of the repetition of the words 'cluster and 'group': ". . .island are clustered into groups. . . clusters are grouped' Do I understand step 4 correctly that the groups are further combined and ordered?

Reply: Thanks for raising this point; the line is indeed confusing and the methods names do not help. We will revise it as follows: "Each step in this methodology is explained further in the following sections: (1) data are first extracted from remotely-sensed imagery and morphometrics are calculated; (2) metrics are normalized and correlation between metrics is addressed with a principal component analysis; (3) island clusters are identified; and (4) clusters are grouped and ordered based on a nested, or hierarchical, clustering scheme."

(R2.8) P6 L6: I'm not sure I agree with this statement as it is written – can you provide a citation for this?

Reply: Fair point, thanks for catching that. We have decided to remove the statement given that it was also placed in the middle of the PCA explanation. The text will be revised as follows: "PCA is a dimensionality reduction tool that preserves the variance within the data while eliminating colinearity between features. PCA reduces this dimensionality, while preserving 90% of the variance in the original dataset."

(R2.9) P7 L3: ClusterPy should be cited look at the 'readme.md' on the github page for the details: https://github.com/clusterpy/clusterpy Also – if there are any other software packages you used i would cite them (software is often not cited in text, but should be).

Reply: Thank you for the suggestion. We have edited the readme file to include all required software packages. The code has been archived and released through Zenodo. The Zenodo DOI will be added to the Code and Data Availability Section. The citation for ClusterPy software package will be added to the text.

Duque, J.C.; Dev, Boris; Betancourt, A.; Franco, J.L. (2011).ClusterPy: Library of spatially constrained clustering algorithms, Version 0.9.9. RiSE-group (Research in Spatial Economics). EAFIT University. http://www.rise-group.org.

(R2.10) P6 and P7, generally: are there any subjective hyperparameters that you set? thresholds for the clustering algorithms? If so, please point those out or mention them.

Reply: Thank you for the comment; we do not have hyperparameters set except those already mentioned in the paper. The main imposed limit was a connectivity constraint, ensuring that only adjacent nodes in the GeoSOM grid could be grouped together.

(R2.11) P8 L 20: To me, figure 4 displays the results of the correlation analysis, not the results from the PCA.

Reply: Thank you for catching that; you are correct and we will edit the text as follows: "The correlation analysis of the normalized metrics (Fig. 4) shows that dimensional

metrics that describe island area show a lesser correlation to factors that scale with area (e.g., dry shape factor, number of outflow channels per island, fractal dimen-sion), whereas those factors are strongly correlated with each other."

(R2.12) P9 L9: I suggest removing the scare quotes on 'sameness', since the dissimi-larity metric is discussed in sect. 3.4.

Reply: We will remove the quotes.

(R2.13) P12 L4: I think the word 'proposed' can be deleted.

Reply: Thanks for the comment; we decided to keep it as if we remove 'here proposed' we need to refer to the analysis as 'this analysis' and it may not be clear if we are referring to the current classification or the one we previously proposed.

(R2.14) Data and Software repository: I see this on github https://github.com/csdmscontrib/DeltaClassification. I would recommend a 'Readme.md' file to help explain the code to make it reproduceable/ extensible for future work. i.e., telling folks where to find the data (i see some '.shp' files in the '_input' folder), and how they could use the code.. I would also provide a DOI for the code itself (via the Zenodo integration with Github) and provide the citation for the code in this manuscript.

Reply: Thank you for the great suggestion. We have packaged the code and created a release in Zenodo. A README file was created and loaded into the repository. This README file contains the required software packages, detailed information about the inputs required to run the code, and an explanation about the code methods. We will provide the link to the repository in the revised manuscript.

(R2.15) Figure 1: I cannot tell the difference between all of the white outlines.

Reply: Thank you, we will revise this figure; this point was also raised by Reviewer 1.